# Bismuth Sulfide Doped in Graphitic Carbon Nitride Degrades Nitric Oxide under Solar Irradiation

**DOI:** 10.3390/nano12193482

**Published:** 2022-10-05

**Authors:** Adnan Hussain, Chitsan Lin, Nicholas Kiprotich Cheruiyot, Wen-Yen Huang, Kuen-Song Lin, Abrar Hussain

**Affiliations:** 1Institute of Aquatic Science and Technology, National Kaohsiung University of Science and Technology, Kaohsiung 811213, Taiwan; 2Ph.D. Program in Maritime Science and Technology, College of Maritime, National Kaohsiung University of Science and Technology, Kaohsiung 81157, Taiwan; 3Department of Marine Environmental Engineering, National Kaohsiung University of Science and Technology, Kaohsiung 81157, Taiwan; 4Super Micro Mass Research and Technology Center, Cheng Shiu University, Kaohsiung City 8333031, Taiwan; 5Center for Environmental Toxin and Emerging-Contaminant Research, Cheng Shiu University, Kaohsiung City 8333031, Taiwan; 6Department of Chemical Engineering and Materials Science, Environmental Technology Research Center, Yuan Ze University, Taoyuan City 32003, Taiwan

**Keywords:** air pollution, heterojunction photocatalyst, solar light degradation, thermal decomposition

## Abstract

This study developed and examined the application of bismuth sulfide doped on graphitic carbon nitride (Bi_2_S_3_@g-C_3_N_4_) in the degradation of NO under solar irradiation. Bi_2_S_3_@g-C_3_N_4_ was prepared through the calcination method. The morphological structure and chemical properties of the synthesized photocatalyst were analyzed before the degradation tests. After doping with Bi_2_S_3_@g-C_3_N_4_, the bandgap was reduced to 2.76 eV, which increased the absorption of solar light. As a result, the Bi_2_S_3_@g-C_3_N_4_ achieved higher NO degradation (55%) compared to pure Bi_2_S_3_ (35%) and g-C_3_N_4_ (45%). The trapping test revealed that the electrons were the primary species responsible for most of the NO degradation. The photocatalyst was stable under repeated solar irradiation, maintaining degradation efficiencies of 50% after five consecutive recycling tests. The present work offers strong evidence that Bi_2_S_3_@g-C_3_N_4_ is a stable and efficient catalyst for the photocatalytic oxidation of NO over solar irradiation.

## 1. Introduction

Air pollutants that are highly reactive include nitrogen oxides (NO_x_), usually caused by anthropogenic activities, particularly fuel combustion [1]. There are two approaches to controlling NO_x_ emissions: primary measures to prevent NO_x_ formation and secondary measures to reduce the NO_x_ already formed [2]. Common primary measures include staged combustion [3] and exhaust gas recirculation [4], while secondary measures include the adoption of pollution control devices including selective catalytic reduction [5] and photocatalysts [6].

Photocatalytic oxidation (PCO), one of the most effective and easiest secondary control technologies for the removal of NO_x_ emissions, has been researched in the last twenty years [7,8,9]. Titanium dioxide (TiO_2_) has been frequently used as a photocatalyst in NO degradation [10] because of its long-term durability, non-toxicity, PCO activity, and long-term photostability [11,12]. However, it has several limitations including a large band gap of 3.0–3.2 eV, and it is only effective under UV light at a wavelength of <380 nm [13,14].

The graphitic carbon nitride (g-C_3_N_4_) semiconductor is a polymeric narrow-band-gap metal-free material that is functional under solar light and has gained increasing popularity [15,16] because of its unique characteristics including high thermal stability, low cost, optical properties, easy preparation, and electron transfer capability [17]. However, in g- C_3_N_4_, electrons and holes easily recombine with each other resulting in low photocatalytic activity [18]. The problem has been addressed through doping with: metals, e.g., Au [19], Fe [20], Pd [21], and Ag [22]; nonmetals, such as P [23], I [24], B [25], and S [26]; and heterojunction composites, e.g., BiPO_4_ and Bi_5_Nb_3_O_15_ [27,28].

This study investigates the outcome of doping bismuth sulfide (Bi_2_S_3_) into g-C_3_N_4_ and the application of the resultant photocatalyst, Bi_2_S_3_@g-C_3_N_4_, on the degradation of NO under solar irradiation. Bi_2_S_3_ is among the sulfide semiconductors with a characteristically large absorption coefficient and narrow bandgap energy [29,30,31]. This study shows that this new photocatalytic material has the potential for application in the degradation of NO in both indoor and outdoor environments.

## 2. Materials and Methods

### 2.1. Chemicals

The lab water purification system (ELGA Lab Water, UK) provided deionized water, bismuth (III) nitrate pentahydrate (Bi(NO_3_)_3_·5H_2_O) was obtained from FERAK^®^ (Berlin, Germany), thiourea 99% (CH_4_N_2_S) and urea (CH_4_N_2_O) were obtained from Alfa Aesar (Lancaster, UK), and the Ming Yang special gas company supplied nitric oxide (100 ppm) compressed gas. All reagents were analytical grade.

### 2.2. Synthesis of Photocatalysts

#### 2.2.1. Synthesis of Bi_2_S_3_

Bismuth sulfide (Bi_2_S_3_) was prepared by the calcination method. First, 0.2512 g of CH_4_N_2_S and 0.9711 g of Bi(NO_3_)_3_·5H_2_O were placed in a mortar and crushed by a pestle for 30 min; then, the mixture was calcinated in a muffle furnace for 2 h at 550 °C. At room temperature, after naturally cooling down, the resulting material was washed 5 times with purified water and ethyl alcohol to eliminate impurities and then dried for 3 h at 80 °C in the oven [32].

#### 2.2.2. Synthesis of g-C_3_N_4_ and Bi_2_S_3_@g-C_3_N_4_

In order to prepare pure g-C_3_N_4_, we mixed and crushed 3.0121 g of CH_4_N_2_S and 10.0113 g of CH_4_N_2_O in a mortar for 30 min and calcinated the mixture in a muffled furnace oven for 2 h at 550 °C. The same procedure was followed while preparing Bi_2_S_3_@g-C_3_N_4_ by the addition of 0.4851 g of Bi(NO_3_)_3_·5H_2_O.

### 2.3. Characterization Procedure

Analyses of the crystallinity and composition of the synthesized materials were performed using X-ray diffractometers (Panalytical Empyrean diffractometer) under Cu Kα radiation (K = 1.5418 Å). A transmission electron microscope was used to inspect the morphological observations and structures (JEOL, JEM-2000FXII), along with a 10 kV scanning electron microscope (SEM). Fourier-transform infrared spectroscopy (FTIR) (JASCO-4700) was utilized to examine the chemical binding variations. The chemical or analytical and optical properties were determined by X-ray photoelectron spectroscopy (XPS) (Thermo Fisher Scientific, Waltham, MA, USA) using K-Alpha and energy dispersive X-ray spectrometry (EDS) (JEOL S4800), respectively.

### 2.4. Experiments Evaluating Photocatalytic Activity

#### 2.4.1. Experimental Procedure

Figure 1 shows an illustration of the lab-scale experimental setup for the NO degradation. Under solar irradiation (λ > 300 nm), using a 300W Xenon lamp as the light source, the synthesized photocatalysts were used to study NO degradation by photocatalysis at ppb levels. The photocatalysts were placed in a 4.5 L stainless steel reaction chamber (10 cm × 15 cm × 30 cm) with a glass on top to allow light to reach the surface of the material. Above the reactor, the lamp was vertically positioned at a distance of 40 cm. The initial NO concentration (500 ppb) was adjusted using air supplied by a zero-air generator, and in the reaction chamber, the gas flowed at a rate of 3 L min^−1^ at 70% relative humidity before flowing into the reaction chamber. By streaming gas into the reaction chamber without light, an equilibrium was achieved between adsorption and desorption. After equilibrium, the degradation experiment started by switching on the light source. The initial and final concentrations of NO were determined by an NO_x_ analyzer (Thermo Scientific, Model 42i). The photocatalytic activity was finished when the minimum level of NO concentration was achieved.

#### 2.4.2. Photocatalytic Activity

In each experiment, 0.2 g of the prepared photocatalyst was dispersed into 20 mL of deionized water in a glass dish (d = 12 cm), rinsed for 5 min in an ultrasonic cleaning bath, and dried for 1 h at 60 °C to completely remove the water. The dishes containing the photocatalyst were placed at room temperature for cooling before being used in the NO degradation experiments. The degradation efficiency of the photocatalytic activity was evaluated by Equation (1):(1)$$\eta \%=\frac{\left(C_{0}-C\right)}{C}\times 100$$
where *η* denotes the degradation efficiency, and *C*_0_ and *C* denote the initial and final concentrations of NO with respect to time.

#### 2.4.3. Evaluation of Active Species

To identify the effective scavengers, trapping tests were conducted by introducing trapping scavengers such as K_2_Cr_2_O_7_, IPA, and KI to trap electrons (e−), hydroxyl radicals (OH), and holes (h+), respectively. The samples were dispersed in a glass dish (d = 12 cm) containing 20 mL of deionized water and the addition of 1%wt of the trapping agent and then sonicated for 20 min. The coated dishes were baked in an oven for 1 h at 60 °C in order to completely remove the water before the NO removal test was conducted.

## 3. Results and Discussion

### 3.1. Characterizations

A TEM and SEM study characterized the morphologies of the pure Bi_2_S_3_, g-C_3_N_4_, and Bi_2_S_3_@g-C_3_N_4_. The TEM images of the synthesized photocatalyst are shown in Figure 2a–c. Figure 2a shows the Bi_2_S_3_ nanostructure composed of nanorods, and Figure 2b shows pure g-C_3_N_4_ with an irregular shape and aggregated layers, but the surface of the flat layers was smooth. The TEM images of composites were investigated to explore the morphology and the structural combination of the photocatalysts. Figure 2c displays the TEM result of the prepared Bi_2_S_3_@g-C_3_N_4_ morphological structure. After calcination of the Bi_2_S_3_@g-C_3_N_4_, the Bi*_2_*S*_3_* nanorods appeared to be well distributed and narrow, which appeared darker than the g-C_3_N_4_ because of its weighted atoms, which were well scattered on the g-C_3_N_4_ with a junction and appeared as a rough surface morphology. Further, a thin layer of the flexible g-C_3_N_4_ nanosheets covered the Bi_2_S_3_ nanorods, which gave rise to a core-shell structure formed between the two composites. As well as wrapping and covering Bi_2_S_3_ nanorods, the g-C_3_N_4_ nanosheets filled the space between them, thus greatly improving the linkage between composites. This unique structure made the Bi_2_S_3_ nanorods more efficient in electron transportation. The resulting images of the Bi_2_S_3_@g-C_3_N_4_ showed the internal structure of the photocatalyst and the presence of both composites, g-C_3_N_4_ and Bi_2_S_3_.

The SEM images identified the Bi_2_S_3_ nanorods over the g-C_3_N_4_, as shown in Figure 2d–f. As shown in Figure 2d, the surface of Bi_2_S_3_ revealed the number of nanorods on its surface with apparent aggregation. As shown in Figure 2e, the g-C_3_N_4_ was established by several typical folded nanosheets, which could be easily analyzed by its edge. Figure 2f shows an SEM image of the Bi_2_S_3_@g-C_3_N_4_, where evidence of holes and folded sheets were visible on its surface. The nanorods of the Bi_2_S_3_ dispersed on the g-C_3_N_4_ folded sheets consistently, which provided a greater surface area, enhanced the porous properties, and provided a greater surface permeability. As a result of these properties, the composite Bi_2_S_3_@g-C_3_N_4_ sample provided an effectively higher capacity for storing charge. Dramatically, the nanorods of Bi_2_S_3_ became narrower than before due to the strong interconnection. These results supported the successful synthesis of the photocatalyst.

The XRD patterns of the synthesized composite before the photocatalytic test are shown in Figure 3a. The diffraction patterns of the composite g-C_3_N_4_ at 2θ = 13° and 27° were well-indexed to the (100) and (002) planes for graphitic materials, respectively. It is common to find two diffraction peaks in all sulfur-based g-C_3_N_4_ products prepared by addition of thiourea [33]. The composite g-C_3_N_4_ showed the two diffraction peaks at 2θ = 13° and 27° were well-indexed to the (1 0 0) and (0 0 2) for graphitic planes, respectively. The observed peak position at 2θ = 13° was associated with the tri-s-triazine in-plane structural unit, and the other observed strong peak at 2θ = 27° was ascribed to the interplanar stacking of the conjugated aromatic system [34], a cyclic phase with the bonds that enhanced the stability and capability as compared to other arrangements. The g-C_3_N_4_ and Bi_2_S_3_ diffraction peaks were noticed in all the Bi_2_S_3_@g-C_3_N_4_ composites, as shown in Figure 3a. There were specific peaks of Bi_2_S_3_@g-C_3_N_4_ observed on the photocatalyst surface; diffraction peaks were observed at 23.1°, 27°, 12.5°, and 33.2°, corresponding to the (013), (143), (121), and (122) planes of the Bi_2_S_3_@g-C_3_N_4_ respectively. Both composites displayed strong and sharp diffraction peaks, which indicated the advantageous crystallinity. By adding Bi_2_S_3_, the intensity of the g-C_3_N_4_ diffraction peaks decreased, while the intensity of the Bi_2_S_3_ increased, demonstrating that the Bi_2_S_3_ was successfully fabricated in the heterojunction with g-C_3_N_4_. Figure 3b shows the XRD pattern of the Bi_2_S_3_@g-C_3_N_4_ composite after the cycle of photocatalytic reactions, where no other new diffraction peaks were produced. This demonstrated that the structure of the Bi_2_S_3_@g-C_3_N_4_ composite did not change the crystallinity. These results indicated that the Bi_2_S_3_@g-C_3_N_4_ composite had good reusability and stability. Further, there was no evidence of impurities, which proved the crystallinity and purity of the photocatalysts.

The Fourier Transform Infrared (FTIR) spectra of the synthesized composites are illustrated in Figure 3c; their surface bonding and chemical groups were examined. The spectrum of pure g-C_3_N_4_ showed absorptions bands between 1150 and 1750 cm^−1^ and 3250 and 3600 cm^−1^. The absorption band of pure g-C_3_N_4_ at 1620 cm^−1^ and 3400 cm^−1^ represented the bending and stretching vibrations of the adsorbed C-N bonds’ and O-H bonds’ residuals, respectively, while the sharp band between 750 and 800 cm^−1^ represented the characteristics of the triazine units. The Bi_2_S_3_@g-C_3_N_4_ and g-C_3_N_4_ did not exhibit any additional peaks in the FTIR spectrum, which indicated a purely physical interface existed between the g-C_3_N_4_ and Bi_2_S_3_.

The elemental mapping technique was used in order to assess the presence of the elements in the Bi_2_S_3_@g-C_3_N_4_. As shown in Figure 4a, by analyzing the EDS spectrum of the photocatalyst Bi_2_S_3_@g-C_3_N_4_, the chemical composition of the prepared nanoparticles was identified, and the peaks of the catalysts Bi_2_S_3_ and g-C_3_N_4_ were examined. The construction of the achieved nanoparticles consisted of Bi, S, C, and N elements. Bi was the dominant element in this catalyst, and elements C and N were also settled in the catalyst. According to the EDS analysis of the synthesized Bi_2_S_3_@g-C_3_N_4_, the ratios of the N, C, S, and Bi for this compound were almost 1:6:13:80. The EDS analysis of the solid-state synthesized Bi_2_S_3_@g-C_3_N_4_ showed that the Bi, S, C, and N ratios for the Bi_2_S_3_@g-C_3_N_4_ were almost 80:13:6:1, respectively. The loading of the Bi in the Bi_2_S_3_@g-C_3_N_4_ was the highest, while the loading of the N and C in the Bi_2_S_3_@g-C_3_N_4_ was the lowest, which affirmed that the synthesized composite was fabricated by both Bi_2_S_3_ and g-C_3_N_4_, which displayed a notable variation between the element content and the input material ratio in the sample.

Furthermore, Figure 4b–e displays the valence state and the chemical compositions of the synthesized material, which were inspected by X-ray photoelectron spectroscopy (XPS). Figure 4b shows the survey and high resolution XPS spectra of the observed elemental signals of the Bi, S, C, and N in the binary system. The elements in the composite Bi_2_S_3_@g-C_3_N_4_ elaborated into Gaussian–Lorentzian peaks. Figure 4c shows the N 1s spectrum of the Bi_2_S_3_@g-C_3_N_4_ presented two obvious peaks at 396.72 eV and 398.34 eV, which were attributed to the sp*^2^* hybridized N atoms involved in the form of C–N–C and the tertiary bridging nitrogen atoms in (N–(C)*_3_*) due to the junction between the g-C_3_N_4_ and Bi_2_S_3_. Figure 4d shows the C 1s spectrum, which presented two peaks found at 282.4 eV and 286.8 eV associated with the sp*^2^* hybridized of C―C and the sp*^2^* hybridized of (N–C=N) bonds in the carbon graphitic structure; in Figure 4e, the high resolution XPS spectrum of Bi 4f displayed two distinct peaks approximately at 157.15 eV and 162.47 eV associated with Bi 4f7/2 and Bi 4f5/2, respectively [35], which showed the evidence of the Bi_2_S_3_ on the g-C_3_N_4_ nanosheets. So, a strong electronic operation was found between the Bi_2_S_3_ and g-C_3_N_4_; the strong interconnection at the interface caused the electrons to move from the Bi_2_S_3_ to the g-C_3_N_4_.

### 3.2. Photocatalytic Test

As shown in Figure 5a, the NO was photocatalytically degraded by the Bi_2_S_3_@g-C_3_N_4_, g-C_3_N_4_, and Bi_2_S_3_ for 30 min under solar irradiation. Due to their large surface areas and low bandgap energies, nanocomposites can display photocatalytic activity. The initial concentration of NO dropped rapidly within 5 min of the photocatalytic activity and eventually reached the minimum concentration over the rest of experiment. The degradation rate of the NO over the photocatalyst Bi_2_S_3_@g-C_3_N_4_ was 55%, which was significantly higher than the Bi_2_S_3_ (35%) and g-C_3_N_4_ (45%), respectively. This was because doping on the g-C_3_N_4_ with Bi_2_S_3_ improved the separation rate between the electrons and holes due to its high absorption capacity. The apparent quantum efficiency (AQE), shown in Figure 5b, is the measurement to compare the utilized photogenerated electrons when the solar light was applied during the photocatalytic degradation of the NO over Bi_2_S_3_@g-C_3_N_4_. However, the degradation efficiency was more than 30% for all the composites, and the AQE was less than 10 × 10^−4^ %, which meant a large number of photons were not operated in this photocatalytic reaction. Figure 5c shows the stability of the photocatalyst Bi_2_S_3_@g-C_3_N_4_ under solar repeated irradiation, studied for its functional operation. A practicable composite sustained the performance in such a way that the photocatalyst could be used several times. To examine the stability, five repetitions of consecutive photocatalytic activities were conducted for the degradation of NO; after the completion of five successive cycles, the photocatalytic degradation of the nitric oxide remained same, and it showed the stability and ability of the catalyst to degrade the nitric oxide under solar light. Moreover, Figure 5d shows the trapping test conducted under solar light to investigate the mechanism of the photocatalytic activity. The photocatalytic degradation of the NO on the materialized catalyst was primarily driven by electrons (e−), hydroxyl radicals (OH), and hole pairs (h+) [36]. The scavengers (KI), (IPA), and (Kr_2_Cr_2_O_7_) were used in this study to trap holes (h+), hydroxyl radicals (OH), and electrons (e−), respectively, in order to examine the active radical species. Furthermore, the electrons (e−) were inhibited significantly when the scavengers were added by 1 wt% to the photocatalyst; these results revealed that photoexcited electrons were the most significant active species undergoing NO degradation.

These results showed that the photogenerated electrons were the primary effective species associated with the photocatalytic reaction. These reactions revealed that the photocatalytic degradation of the NO over the Bi_2_S_3_@g-C_3_N_4_ was primarily driven by the photogenerated electrons and holes.

### 3.3. Photocatalytic Mechanism of NO Degradation

Figure 6 shows the optical analysis of the prepared composites. The band gap of the synthesized photocatalyst is shown in Figure 6a, which defined the electric conductivity of the photocatalysts. The Tauc method was introduced to determine the band gap of the photocatalyst. Based on the relationship between the Eg and the optical absorption, the band gap energies of the Bi_2_S_3_, g-C_3_N_4_, and Bi_2_S_3_@g-C_3_N_4_ were evaluated. A wider band gap is reduced through the doping of metal oxides. The band gap of the Bi_2_S_3_@g-C_3_N_4_, observed as 2.76 eV, was the gap between the bonding molecular orbitals and the antibonding molecular orbitals. The doping process, by adding Bi_2_S_3_ and g-C_3_N_4_, altered the Fermi level, replaced the CB in g-C_3_N_4_, and replaced the VB in Bi_2_S_3_, which narrowed the distance between the CB and VB. In addition, these dopant atom orbitals acted as intermediate states or bridged the gap between the valence and conduction band of electrons of the Bi_2_S_3_ and g-C_3_N_4_, some new bonds were formed, and the number of molecular orbitals increased resulting in shorter energy for electronic transitions (a lower bandgap). Furthermore, a smaller particle size led to enhanced electron transfer properties, preventing electron–hole recombination and reducing the band gap of the composite, thereby increasing the photocatalytic activity. When the band gap was reduced, excitation occurred at lower irradiation powers. The determined energy band gap was estimated as 2.86 eV and 3.1 eV for the g-C_3_N_4_ and Bi_2_S_3_, respectively, while for the Bi_2_S_3_@g-C_3_N_4_, it was estimated as 2.76 eV. Figure 6a shows the observed band gaps as compared to the Bi_2_S_3_ and g-C_3_N_4_; the Bi_2_S_3_@g-C_3_N_4_ showed an improvement in light absorption, electron level structure, surface area, particle size, and morphology. The Eg of the Bi_2_S_3_@g-C_3_N_4_ corresponded less to the Bi_2_S_3_ and g-C_3_N_4_, which showed the potential for photocatalytic activity. The nanojunction of the Bi_2_S_3_ and g-C_3_N_4_ was advantageous for utilizing and absorbing the solar light, which was the cause of the formation of more electrons and performed as a good photocatalyst. For the photocatalytic mechanism of the prepared heterojunction composites shaped by the junction of the Bi_2_S_3_ and g-C_3_N_4_ shown in Figure 6b, both composites contributed a supportive platform to carry the photogenerated charge carriers from the valence band (VB) to the conduction band (CB) of the Bi_2_S_3_ and g-C_3_N_4_. When the Bi_2_S_3_ and g-C_3_N_4_ became excited under solar light, a photoinduced electron–hole pair was generated. In the Bi_2_S_3_, the electrons at the CB recombined rapidly with the holes at the VB of the g-C_3_N_4_ due to the synergistic effects of an inner electric field, Coulomb interaction, and band bending; during this time, the functional electrons and holes remained attached to the CB and VB of the g-C_3_N_4_ and the Bi_2_S_3_, respectively, for participation in the redox reaction. Based on the result, the charge carriers were efficiently separated and utilized, and the heterojunction photocatalyst boosted the photocatalytic activity, which was notably improved.

## 4. Conclusions

The present study provided an effective method for preparing a photocatalyst for the degradation of NO under solar light based on Bi_2_S_3_@g-C_3_N_4_. The degradation efficiency of the NO by the Bi_2_S_3_@g-C_3_N_4_ achieved 55%. The photocatalytic activity of the Bi_2_S_3_@g-C_3_N_4_ was attributed to the synergistic effects that occurred between the two heterojunction composites of g-C_3_N_4_ and Bi_2_S_3_, which expanded the absorption of solar light and accelerated the electron–hole pair separation efficiency. In the future, this photocatalyst could be tested on other pollutants, such as volatile organic compounds, under solar or visible light or coated on the surface of tiles and bricks and used in indoor and outdoor environments for the degradation of atmospheric pollutants.

## Figures and Tables

**Figure 1 nanomaterials-12-03482-f001:**
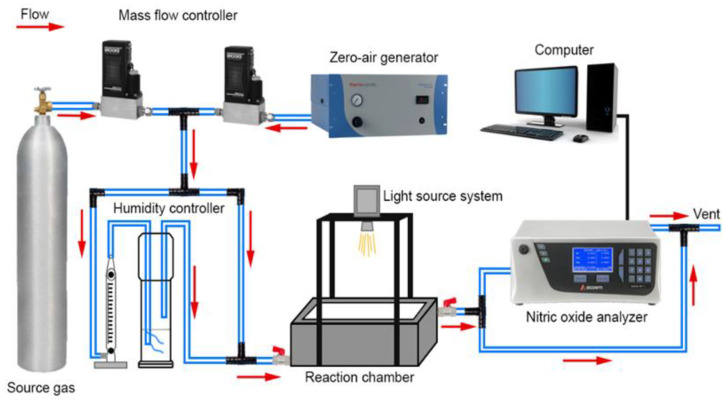
Illustration of the experimental layout in this study.

**Figure 2 nanomaterials-12-03482-f002:**
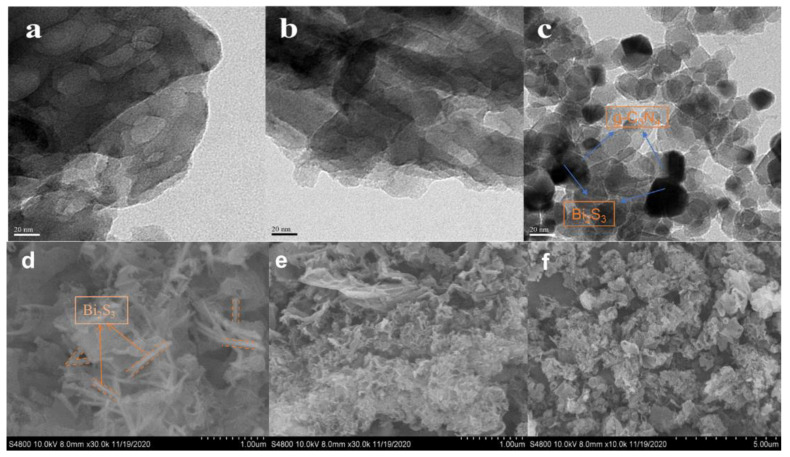
The TEM images of the synthesized (**a**) Bi_2_S_3_, (**b**) g-C_3_N_4_, and (**c**) Bi_2_S_3_@g-C_3_N_4_. The SEM images of the synthesized (**d**) Bi_2_S_3_, (**e**) g-C_3_N_4_, and (**f**) Bi_2_S_3_@g-C_3_N_4_.

**Figure 3 nanomaterials-12-03482-f003:**
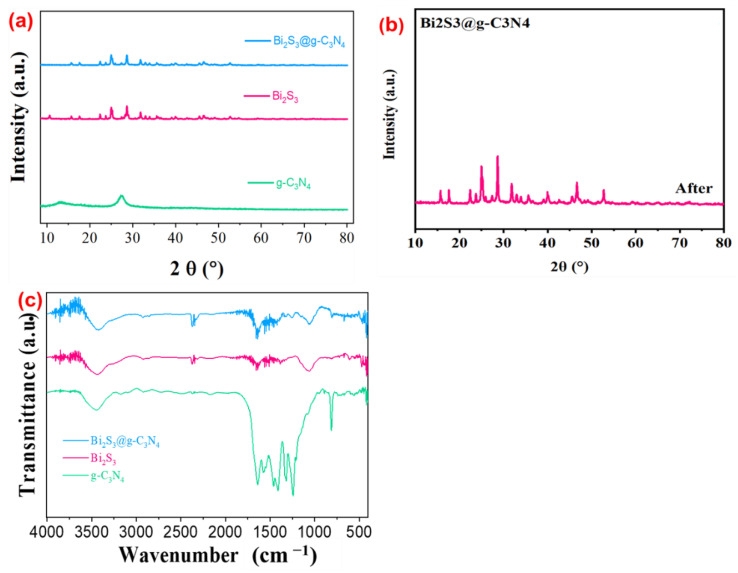
XRD and FTIR analysis results. (**a**) The XRD patterns of the synthesized Bi_2_S_3_@g-C_3_N_4_, Bi_2_S_3_, and g-C_3_N_4_ before the photocatalytic test. (**b**) The XRD analysis of the synthesized Bi_2_S_3_@g-C_3_N_4_ after the photocatalytic test and the (**c**) FTIR spectra analysis of the prepared Bi_2_S_3_@g-C_3_N_4_, Bi_2_S_3_, and g-C_3_N_4_.

**Figure 4 nanomaterials-12-03482-f004:**
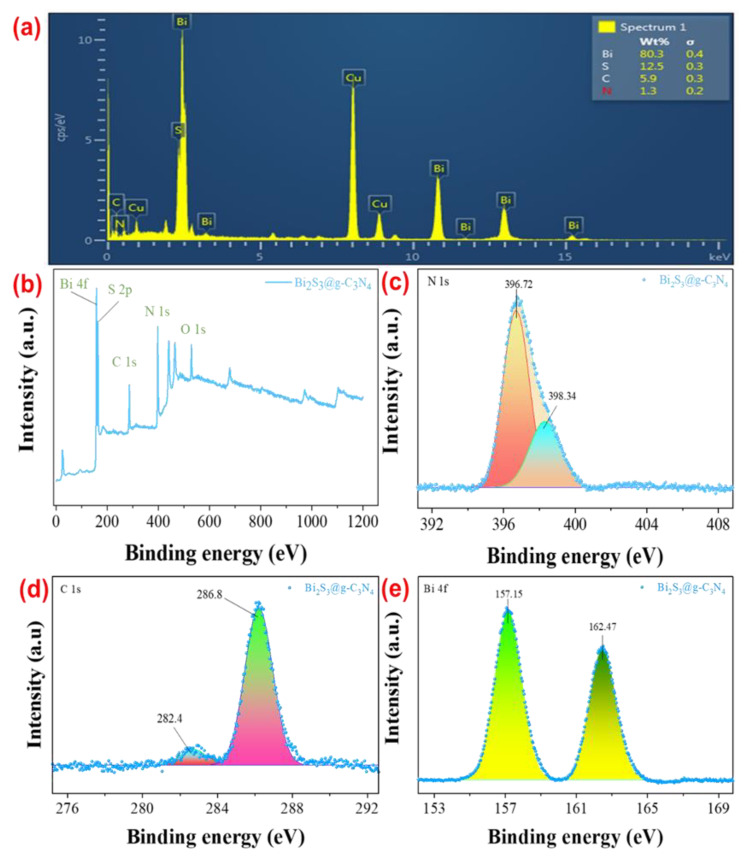
(**a**) The EDX-mapping of the synthesized Bi_2_S_3_@g-C_3_N_4_, (**b**) the XPS Survey, and the XPS high resolution spectra, (**c**) the N 1s spectrum, (**d**) the C 1s spectrum, and (**e**) the Bi 4f spectrum.

**Figure 5 nanomaterials-12-03482-f005:**
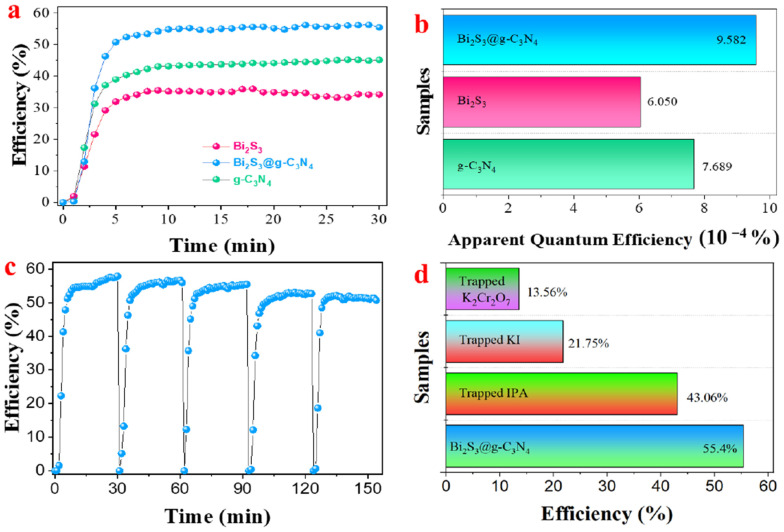
(**a**) The photocatalytic degradation, (**b**) apparent quantum efficiency, (**c**) recycling test, and (**d**) trapping test of the synthesized material under solar irradiation.

**Figure 6 nanomaterials-12-03482-f006:**
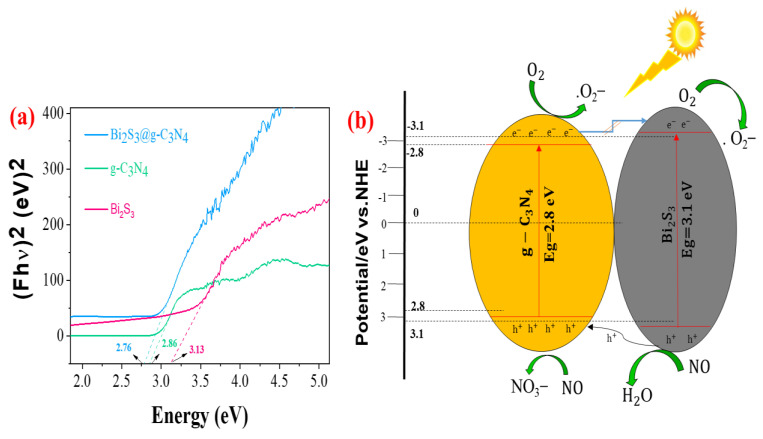
(**a**) Tauc plots of the synthesized materials and (**b**) the photocatalytic proposed mechanism of the Bi_2_S_3_@g-C_3_N_4_ under solar irradiation.

## Data Availability

Not applicable.

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
