# Peer review of "Bismuth Sulfide Doped in Graphitic Carbon Nitride Degrades Nitric Oxide under Solar Irradiation"

_nanomaterials, 2022, doi:10.3390/nano12193482_

Round 1
Reviewer 1 Report
The manuscript "Bismuth sulfide doped in graphitic carbon nitride degrades nitric oxide under solar irradiation" investigates the application of bismuth sulfide doped on graphitic carbon nitride (Bi2S3@g-C3N4) in the degradation of NO under solar irradiation. It is shown that the bandgap of the doped Bi2S3@g-C3N4 structure reduces to 2.76 eV, which increased the absorption of solar light. As a result, Bi2S3@g-C3N4 achieved higher NO degradation (55%) compared to pure Bi2S3 (35%) and g-C3N4 (45%). The work claims the Bi2S3@g-C3N4 structure is stable and efficient catalyst for photocatalytic oxidation of NO over solar irradiation. The work presents scintific interest and deseve to be published after the following comments will be addressed.
Comment 1. The manuscript should be profread. For example, lines 36-37, the Authors forgot to remove part of the template's text "The introduction should briefly place the study in a broad context and highlight why it is important. It should define the purpose of the work and its significance."
Comment 2. In Section 3.3 Photocatalytic mechanism of NO degradation, the band gap sizes of Bi2S3, g-C3N4, and Bi2S3@g-C3N4 are discussed. It is shown that the band gap of Bi2S3@g-C3N4 decreases compared to these of Bi2S3 and g-C3N4. It is interesting to discus a mechanism of this change.
Author Response
Response to Reviewer 1 Comments
Point 1: The manuscript should be proofread. For example, lines 36-37, the Authors forgot to remove part of the template's text "The introduction should briefly place the study in a broad context and highlight why it is important. It should define the purpose of the work and its significance."
Author Response: The authors are truly thankful to the respected reviewer for his/her effort and time in reviewing the manuscript and the positive assessment of the work. Regarding reviewer comment, the part of templet text is removed and updated the manuscript, please see the lines 39 – 40. Moreover, the paper is carefully proofreaded.
Point 2: In Section 3.3 Photocatalytic mechanism of NO degradation, the band gap sizes of Bi2S3, g-C3N4, and Bi2S3@g-C3N4 are discussed. It is shown that the band gap of Bi2S3@g-C3N4 decreases compared to these of Bi2S3 and g-C3N4. It is interesting to discuss a mechanism of this change.
Author Response: Authors thanks the reviewer for this valuable suggestion. Following your comment, we have updated the section 3.3 photocatalytic mechanism of NO by adding a new text from line 274-289. Please see the newly added part below:
The band gap of Bi2S3@g-C3N4 observed as 2.76 eV is the gap between bonding molecular orbitals and antibonding molecular orbitals. The doping process by adding Bi2S3 and g-C3N4 alters the Fermi level and replaces the CB in g-C3N4 and replaces the VB in Bi2S3 which could narrow the distance between CB and VB. In addition, these dopant atom orbitals act as intermediate state or bridge the gap between valence and conduction band of electrons of the Bi2S3 and g-C3N4, some new bonds are formed and the number of molecular orbitals were increased resulting shorter energy for electronic transitions (lower bandgap).

Reviewer 2 Report
Here are my comments on the manuscript titled “Bismuth sulfide doped in graphitic carbon nitride degrades nitric oxide under solar irradiation”.
Decision: Major revision
Hussain et al. developed Bi2S3/g-C3N4 for photocatalytic degradation of NO. The authors investigated the morphological structure and chemical properties. The as-synthesized Bi2S3@g-C3N4 was used to investigate the photocatalytic performance of NO under the solar light. The authors should solve few major concerns before publishing this paper in Nanomaterials.
1. The authors claimed Bi2S3@g-C3N4 was prepared through the grinding method. However, according to the synthesis of photocatalysts, all samples were prepared through calcination.
2. Does the sulfur atom appear in pure g-C3N4 sample?
3. In Figure 2, it is suggested to show a HRTEM image with clear fingerprint of Bi2S3.
4. Does the introduction of Bi2S3 largely change the surface area of Bi2S3@g-C3N4?
5. Does the amount of bismuth nitrate change the photocatalytic performance?
6. A comparison of XRD for Bi2S3@g-C3N4 before and after photocatalytic test should be conducted.
In addition to the above major concerns, there are also some typos to be corrected such as “1150-1750 cm-1” and “sp2 C C and sp2 (N C N)”. The authors should read the manuscript thoroughly. I list two examples here for correction. 1) The introduction should briefly place the study in a broad context and highlight why it is important. It should define the purpose of the work and its significance. 2) Photocatalytic oxidation (PCO), one of the most effective and easiest secondary control technologies for the removal of NOx emissions researched in the last twenty years [7, 8, 9].
Author Response
Response to Reviewer 2 Comments
Point 1: The authors claimed Bi2S3@g-C3N4 was prepared through the grinding method. However, according to the synthesis of photocatalysts, all samples were prepared through calcination:
Author Response: The authors are truly thankful to the respected reviewer for his/her effort and time in reviewing the manuscript and the positive assessment of the work. Regarding reviewer comment. All the samples were prepared by calcination after grinding the proposed chemicals in the manuscript. For the better understanding I changed the text “prepared by calcination method” instead of “grinding method”. Please see the line 75.
Point 2: Does the sulfur atom appear in pure g-C3N4 sample?
Author Response: We thank the reviewer for this valuable suggestion. Pure g-C3N4 was prepared by mixing of thiourea (CH4N2S) and urea (CH₄N₂O) which is mentioned in 2.2.2. While thiourea is known as an organosulfur compound which contains sulfur. The XRD analysis of pure g-C3N4 confirms the presence of two diffraction peaks. we have updated by adding a new lines in the section 3.1. Please see the added text in line 169-170.
“It is common to find two diffraction peaks in all sulfur-based g-C3N4 products prepared by addition of thiourea”.
The reference added in the subjected section are as below:
[33] An, T. D., Phuc, N. V., Tri, N. N., Phu, H. T., Hung, N. P., & Vo, V. Sulfur-doped g-C3N4 with enhanced visible-light photocatalytic activity. Appl. Mech. Mater. (2019), 889, 43-50.
Point 3: In Figure 2, it is suggested to show a HRTEM image with clear fingerprint of Bi2S3.
Author Response: Thank you for your suggestion to improve the quality of manuscript, the authors have to say unwillingy that we donot have HRTEM image but for better understaing we updated the Figure 2.c TEM analysis of Bi2S3@g-C3N4 to show the decomposition of Bi2S3. The TEM images also clearly reveal a close integration between the g-C3N4 nanosheets and Bi2S3 nanoparticles, which is conducive to the transportation of charge carriers. The relatively small size of the Bi2S3 nanorods and the high surface to volume ratio of the g-C3N4 nanosheets combined, could explain the presence of individual Bi2S3 nanorods surrounded by g-C3N4 shells .It is requested to the reviewer please see the Page 2, Lines 145–150 and updated figure (2.c), Page 2, Lines 162–163.
Point 4: Does the introduction of Bi2S3 largely change the surface area of Bi2S3@g-C3N4?
Author Response: Thank you for your comment, Bi2S3 change the surface area of Bi2S3@g-C3N4 is discussed in the section 3.1, SEM analysis. Please see the line 156-158.
“The nanorods of Bi2S3 dispersed on g-C3N4 folded sheets consistently, which provide a greater surface area, enhance porous properties, and greater surface permeability”. Furthermore, the SEM results show that Bi2S3 nanorods cover on the surface of g-C3N4 nanosheets uniformly. More interestingly, Bi2S3 nanorod becomes slender than before due to the strong interconnection between Bi2S3 and g-C3N4 restrict the aggregation of Bi2S3. Bi2S3 porous layered has a large specific surface area and pores, so it provides more defects with which to facilitate the crystallisation, nucleation and growth of Bi2S3@g-C3N4.
Point 5: Does the amount of bismuth nitrate change the photocatalytic performance?
Author’s Response: The authors thank the reviewer comment. I did not prepare the composite with different mass ratios. 0.4851 g of Bi (NO3)3.5H2O was used to prepare the Bi2S3@g-C3N4 by following previously reported literature which is mentioned as a reference in page 2, “line 80”.
The reference mentioned in the section 2.1 are as below:
[32] Hu, T., Dai, K., Zhang, J., Zhu, G., & Liang, C., One-pot synthesis of step-scheme Bi2S3/porous g-C3N4 heterostructure for enhanced photocatalytic performance. Mater. Lett. 2019, 257, 126740.
Point 6: A comparison of XRD for Bi2S3@g-C3N4 before and after photocatalytic test should be conducted.
Author’s Response: The authors are truly thankful to the reviewers to englithen the basic point of the study. It is important to show the XRD analysis before and after the photocatalytic test to confirm the diffraction peaks and crystallinity of the photocatalyst. XRD analysis of Bi2S3@g-C3N4 after the photocatalytic test is updated in the manuscript named as Figure 3b. The following text is added to compare before and after photocatalytic test of Bi2S3@g-C3N4. Please see the page 5, lines 183 – 187.
Figure 3b. shows the XRD pattern of Bi2S3@g-C3N4 composite after the cycle of photocatalytic reactions, where no other new diffraction peaks are produced. This demonstrates that the structure of the Bi2S3@g-C3N4 composite did not change the crystallinity. These results indicate that the Bi2S3@g-C3N4 composite has good reusability and stability.
We are very grateful for the reviewer’s valuable comments and suggestion. The comments are encouraging and the reviewer appears to share our judgment that this study and the results are of interest to readers. We have improved the manuscript according to the reviewer’s suggestion. Thank you so much once again for the kind reminder and great suggestion of reviewers.

Round 2
Reviewer 2 Report
Here are my comments on the revised manuscript titled “Bismuth sulfide doped in graphitic carbon nitride degrades nitric oxide under solar irradiation”.
Decision: Acceptance
The authors have addressed all concerns. Hence, I suggest accepting it for being published in Nanomaterials.